# Limited Optimal Plastic Behavior of RC Beams Strengthened by Carbon Fiber Polymers Using Reliability-Based Design

**DOI:** 10.3390/polym15030569

**Published:** 2023-01-22

**Authors:** Sarah Khaleel Ibrahim, Majid Movahedi Rad

**Affiliations:** Department of Structural and Geotechnical Engineering, Széchenyi István University, 9026 Gyor, Hungary

**Keywords:** reliability index, complementary strain energy, probability, haunched beams, CFRP, optimum solution

## Abstract

The plastic behavior of strengthened haunched beams utilizing carbon fiber-reinforced polymers (CFRP) was investigated using a probabilistic design that took into account random concrete properties, CFRP properties, and complementary strain energy values, with the reliability index serving as a limiting index, as the proposed method considers a novel method that deals with probabilistic parameters for models with limited plastic behavior designed based on the reliability index. The data used in this research were gathered and evaluated in a recent study on simply supported haunched beams reinforced with carbon fiber-reinforced polymers. The purpose of this research was to use the reliability limitation index for simulated strengthened haunched beams by taking into account randomness in concrete and CFRP properties and the complementary strain energy value, which is considered a plastic behavior controller that provides an illustration of the damage amount within the reinforcement steel bars. The results indicate how randomness affects the behavior of the presented models, which are chosen to have different numbers of CFRP strips. The variable randomness affects load and deflection values where the reliability index value increases as the corresponding load value decrease, reflecting the increased probability of failure in models subjected to higher loading conditions, while tension concrete damage percentages are reflected in the damage pattern presented in the results, showing that as the produced load increases, so does the damage intensity. It is also obvious that the reliability index served as a limitation index while taking concrete characteristics and complementary strain energy as random variables.

## 1. Introduction

Economical solutions are significantly important for structural engineers, and for any solution the safety of the structure should be obtained. Non-prismatic beams, which may also be called as haunched beams, hold an optimal shape that maximizes the beam section at the support area while keeping it at the minimum allowed in the mid-span. This haunched shape is considered to be more economical if compared with normal prismatic beams at the same provided strength. Therefore, this type of beam has been highlighted by different research over many years. To accomplish these goals while keeping costs down, reinforced concrete haunched beams were studied by Haque [1]. Parametric investigations under gravity loading were performed, and linear and non-linear formulas for 5-noded line elements were obtained. The beam depth was reduced in the parametric studies due to changes in haunch angles and lengths. Most reinforced concrete haunched beams cases, strains, and displacements deviate from the 3° haunch angle, according to parametric research. The non-linear model and solution technique of that research can demonstrate a haunched beams acceptance level due to residual stress generation of the iterative process. López-Chavarrıa et al. [2] determined the uses of mathematics to determine the optimum design of straight-haunched T-section beams made of reinforced concrete by minimizing the cost of concrete and steel while the geometric qualities serve as the constraints. Positive and negative moments at supports, as well as the maximal moment, are all displayed in the study. The study uses numerical comparisons to examine the relative qualities of T-section and rectangular beams, showing that the T-beams are more space-efficient, lighter, and less expensive than the rectangular sections.

Due to the inclusion of heterogenic material characteristics and the cracking behavior of concrete, the non-linear behavior of RC beams up until the ultimate collapse is a complex process. Experimental testing is typically used to estimate the behavior of reinforced concrete elements up until failure, but due to the high expense of testing apparatus and materials, observations are typically only collected at crucial spots. Behavior prediction of RC beams is typically carried out using numerical approaches to eliminate destructive testing, and lower material and labour costs. Therefore, a lot of work has been done on the non-linear FEA of RC beams under different considerations [3,4,5,6].

Moreover, strengthening structures using carbon fiber-reinforced polymers (CFRP) is considered an effective and economical solution in some cases if compared with the demolition solutions. CFRP strips are easily installed in the weak parts, aiming to strengthen them and delay the beginning of micro cracks and prevent their progress. Different research was conducted to prove the efficiency of strengthening methods such as the work presented by Lu et al. [7], which used the pre-stressed CFRP fabric to reinforce RC columns. The results demonstrated that the load-bearing capacity and ductility of RC columns could be greatly enhanced by incorporating CFRP fabric. On the other hand, Alabdulhady et al. [8] investigated the effects of a CFRP strengthening and repair system on RC beams with varying concrete compressive strengths. Flexural load testing was performed on RC beams after they were strengthened and repaired using CFRP composite. Low, medium, and high strength was demonstrated by casting the beams with concrete with different compressive strengths. The concrete compressive strength was inversely related to the performance of the CFRP composite material in repaired beams. Besides, the performance of externally applied wraps on simply supported RC beams under pure torsion was defined by Zhou et al. [9], and their research looks at how eccentric torsional loads combined with bending and shear pressures affect damage in framed beams that are mounted to columns using a variety of wrapping procedures. Wrap performance and failure modes are investigated using verified FE method. The results demonstrated that the flexural capabilities were diminished in direct proportion to the amount of torsional force applied and the concrete strut’s capacity and torsional shear-flow are improved by the CFRP wraps, which are also compatible with the imposed load ratios and the specified truss mechanism. Additionally, rectangular reinforced concrete (RC) columns with carbon fiber-reinforced plastic (FRP) confinement were modelled in three dimensions using a meso-scale approach that takes concrete variability into account by Fan et al. [10]. There was a size effect on the confinement of CFRP jackets on concrete, as measured by a decrease in the effective confinement area ratio estimated from the axial stress distribution as the structure size increased. The effect of confinement’s size diminished with increasing aspect ratio and edge sharpness. 

Concrete buildings can be significantly strengthened by utilizing carbon fiber-reinforced plastic (CFRP) due to this material’s high strength, low weight, great corrosion resistance, creep resistance, and fatigue resistance. Xian et al. [11] used wedge-extrusion bond anchorage for CFRP plate anchorage load-bearing. As a pre-stressed tension device combined high temperature, distilled water, and constant loading, applying a mechanical analysis and tensile tests of the anchorage system showed CFRP burst failure mechanisms under static and cyclic loads without anchor de-bonding. This resulted in the CFRP plate’s homogenous stress distribution parallel to the anchor’s biggest cross-section, which increased anchor-age capacity. In addition, durability statistics and research are needed to understand the deterioration mechanisms of bonded connections between CFRP laminates and steel substrates under harsh environmental conditions. Double-strap CFRP-to-steel bonded connections used two composite materials. Yang et al. [12] examined adhesive coupons to cover the gap where a static tensile machine also replicated the number of temperature cycles and slippage induced in CFRP laminates. Additionally, civil engineering structural strengthening and maintenance are complicated by carbon fiber reinforced polymer (CFRP) composite fire resistance. Temperature exposure affected the mechanical, thermal, and microstructure properties of carbon fibers and CFRP plates with bisphenol-A epoxy matrix and hydantoin epoxy matrix. Thermal breakdown, void content, surface morphology, and internal microstructures revealed CFRP plate deterioration mechanisms [13].

However, optimal solutions were presented and explored to manage the behavior of structures by considering various structures and approaches to produce the best possible results in terms of strength, shape, cost, etc. The optimization method intends to find the best path that any solution can take while defining an objective condition and constraints that control the whole operation. Nassif et al. [14] presented a project that intends to optimize fiber integration for ready-mix, worksite use. A high-performance concrete (HPC) mix incorporating mixed, various fiber types (crimped steel, macro polypropylene, and micro polypropylene) was evaluated. These “hybrid” fiber blends were investigated for mechanical properties and durability. Moreover, Tamrazyan and Alekseytsev [15] looked to optimize reinforced concrete buildings using multiple criteria. The potential for both natural disasters and accidental harm caused by humans is factored into the latest optimality standards. A customized genetic algorithm was developed for use as a search engine where the system can handle a wide variety of accident conditions, such as corrosive damage and local mechanical damage. 

Further, one of the important facts structural engineers face is the uncertainty in the structural characteristic that can affect the behavior of any structure dramatically while influencing the probability of failure, which, in its turn, reflects the reliability of any condition, Zheng et al. [16] used ultrasonography to assess the level of damage to beams that had previously been damaged, repaired, and strengthened with high ductility concrete (HDC), and then gave probabilistic statistical analysis of the results. Hassanzadeh et al. [17] demonstrated a statistical evaluation of how concrete and GFRP rebar acted against one another in terms of binding strength. This probabilistic model provides a quantitative expression for the epistemic uncertainties associated with the parameters and errors of the model, which decrease with additional data. The problem mechanics were used to select the variables and the explanatory functions. All of these factors were considered in order to select the most appropriate model: prediction precision, error non-normality, scattering non-homogeneity, residual correlation, and non-linearity. A study contrasting deterministic versus probabilistic models was held, representing the role of uncertainty. Likewise, measurement uncertainty for deformed steel bars used in concrete reinforcement was evaluated by Suhartono and Rustiant [18], where different diameters of deformed steel bars were considered to calculate the margin of error in the measurements, static tensile testing was performed. The analysis evaluated the measurement uncertainty by collecting more data. Similarly, predicting load-bearing capacity can be modelled using material properties, geometrical parameters, and failure, causing uncertainty as presented by Sykora et al. [19], and investigating the shear resistance model uncertainties for reinforced concrete beams with and without shear reinforcement. Model uncertainty can be characterized through a comparison of test and model outcomes, where the uncertainty of the model is highly sensitive to the parameters.

After highlighting some previous studies, it is worth mentioning that this study collects the models presented by Ibrahim and Rad [20] in 2020, and the collected models are reinforced concrete haunched beams strengthened by CFRP strips. The models were calibrated using Abaqus while using the experimental results obtained by testing beam models and concrete samples in the laboratory. The authors then offered a programming code that could be used to apply the deterministic optimization problem to the numerically calibrated haunched beams, with the complementary strain energy of the internal stresses initiated within the rebar being taken into account as a limiting factor. After that, a second code was given to solve the probabilistic solution for the same strengthened reinforced haunched beams, where the concrete characteristics, complementary strain energy, and CFRP characteristics are all treated as probabilistic values with mean values and standard deviations, while the optimization process is constrained by reliability index values. This research primarily focuses on two strengthening scenarios, where two and three CFRP strips are employed to reinforce the haunched beams in the shear section at 45 degrees angles. The outcomes are displayed, taking into account the two strengthening examples offered for both deterministic and probabilistic solutions. The probability solution provides insight into how the standard deviation of probabilistic values affects the strength, behavior, and damage intensity of the models, while the deterministic case demonstrates how the complementary strain energy bounds the plastic behavior of the beam, which is reflected by the models’ damage behavior. In this research, the uncertainties that the concrete material is subject to are taken into account, which is the actual state of this material, as it is hard to obtain its properties without subjecting it to probability. Given the fact that CFRP is one of the effective materials in strengthening the structural elements, therefore, the uncertainties of the CFRP properties were taken into account to study the effect of these materials probabilities on the behavior of the reinforced concrete beams strengthened, with different numbers of CFRP strips owning variable properties.

After this introduction, Section 2 provides a detailed explanation of the methodology used in this paper. Following this is Section 3, which displays the calibrated models, followed by Section 4, which displays the results and discussions. Finally, Section 5 gives the main conclusions of this study.

## 2. Methodology

### 2.1. Plastic Behavior Limitation Principal

This method is used for plastic analysis and design where the residual stresses exist, as complementary strain energy is successfully used in different types of structures [21,22,23,24], and in this research it is used as a limitation of the failure of the structure.

When such energy amount restrictions are necessary to govern residual deformations [22,23,24], to take into account the strain energy of residual forces as a comprehensive evaluation of plastic behavior, a suitable computational technique was developed. The residual forces that produce this supplementary strain energy are presented:(1)$$W_{p}=\frac{1}{2E}\;\overset{n}{\underset{i=1}{\sum }}\frac{l_{i}}{A_{i}}\;N_{i}^{R^{2}}\leq W_{p0}$$

Here, $W_{p0}$ is the maximum energy that may be used to calculate $W_{p}$ from the structure’s elastic strain energy [22]. The Young’s modulus of the bar material is E, the residual force of the bar members is $\;N_{i}^{R}$, and the length of the bar elements is $l_{i},\;\left(i=1,\;2,\;\ldots ,\;n\right)$, the cross-sectional area of the bar elements is $A_{i},\;\left(i=1,\;2,\;\ldots ,\;n\right)$, and so on. In Equation (1), we see the introduction of a limit value $W_{p0}$ for plastic rebar deformations. Aside from the inner plastic force $\;N^{pl}$, which will occur when the load $P_{0}$ is applied, and the elastic internal force $-N^{el}$, the residual forces $N^{R}$ that are displayed in the structure after unloading are described by these two forces.
(2)$$N^{R}=N^{pl}-N^{el}$$
where:(3)$$N^{el}=F^{-1}G^{T}K^{-1}P_{0}$$

In this context, Matrix F denotes adaptability, Matrix G denotes geometry, and Matrix K denotes rigidity.

### 2.2. Probability Theory

We assume the probability density functions of $X_{R}$ and $X_{S}$ are $f_{R}\;\left(X_{R}\right)$ and $f_{S}\;\left(X_{S}\right)$, respectively, where $X_{R}\leq X_{S}$. It is possible to assess the probability of failure using the following equations [25]:(4)$$P_{f}=P\left[X_{R}\leq X_{S}\right]=\iint_{X_{R}\leq X_{S}}^{}f_{R}\;\left(X_{R}\right)f_{S}\;\left(X_{S}\right)dX_{R}dX_{S}$$

To briefly describe the issue, we can use the well-known bound state function, which is defined as:(5)$$\;\;\;\;\;\;\;g\left(X_{R},X_{S}\right)=X_{R}-X_{S}$$

The failure domain, $D_{f}$, is represented by the value g ≤ 0. Therefore, we can write down the failure probability $P_{f}$:(6)$$P_{f}=F_{g}\left(0\right)$$

Additionally, $P_{f}$ can be given by:(7)$$P_{f}=\int_{g\left(X_{R},X_{S}\right)\leq 0}^{}f\left(X\right)dX=\int_{D_{f}}^{}f\left(X\right)dX$$

The complementary strain energy of the residual forces is constrained in this study by a Gaussian distribution with a mean of ${\bar{W}}_{po\;}$ and a standard deviation of $\sigma_{w}$, reflecting the uncertainty associated with the data.

The probability of failure ($P_{f}$) values are used to calculate the reliability index (β), with the Monte-Carlo sampling method additionally taken into account. Generation of realizations x according to a random vector X of the probability density function $f_{X}\left(x\right)$ according to Monte-Carlo method. Simply counting the total number of points allows one to determine the fraction of points ($P_{f}$) that are in the failure domain. Using a defined indicator function of $D_{f}$, we can write down the following to express this idea.
(8)$$\chi_{D_{f}}\left(x\right)=\left\{\;\begin{array}{c} 1\;\;\;\mathrm{if}x\in \;D_{f}\\ 0\;\;\;\mathrm{if}x\notin \;D_{f} \end{array}\right\}$$

Through rephrasing Equation (7):(9)$$P_{f}=\int_{-\infty }^{+\infty }...\int_{-\infty }^{+\infty }\chi_{D_{f}}\left(x\right)f_{X}\left(x\right)dx$$

Hence χDf(X) is two-points variable distributed randomly.
(10)ℙ[ χDf(X)=1]=Pf  
(11)ℙ[ χDf(X)=0]=1−Pf

That Pf=ℙ[ X∈ Df ].

Mean and standard deviation for a random variable $\chi_{D_{f}}\left(X\right)$ can be calculated as follows:(12)$$E\left[\chi_{D_{f}}\left(X\right)\right]=1\cdot P_{f}+0\cdot \left(1-P_{f}\right)=P_{f}$$
(13)$$Var\left[\chi_{D_{f}}\left(X\right)\right]=E\left[\chi_{D_{f}}^{2}\left(X\right)\right]-{\left(E\left[\chi_{D_{f}}\left(X\right)\right]\right)}^{2}=P_{f}-P_{f}^{2}=P_{f}\left(1-P_{f}\right)$$

To evaluate $P_{f}$ using the Monte-Carlo method, using the next formula:(14)$$\hat{E}\left[\chi_{D_{f}}\left(X\right)\right]=\frac{1}{Z}\overset{Z}{\underset{z=1}{\sum }}\chi_{D_{f}}(X^{\left(z\right)})={\hat{P}}_{f}$$

That $X^{\left(z\right)}$ is a demonstration for a set of independent random vectors with (z=1,…,Z), and $f_{X}\left(x\right)$.

It is worth noting that the complementary strain energy is accounted for as a random variable in probabilistic models. Therefore, we can calculate its mean value and standard deviation. Furthermore, it has a mean of E and a variance of Var, which is consistent with the Gaussian distribution. One may quickly determine the estimator’s mean and standard deviation by using:(15)$$E\left[{\hat{P}}_{f}\right]=\frac{1}{Z}\overset{Z}{\underset{z=1}{\sum }}E\left[\chi_{D_{f}}\left(X^{\left(z\right)}\right)\right]=\frac{1}{Z}ZP_{f}=P_{f}$$
(16)$$Var\left[{\hat{P}}_{f}\right]=\frac{1}{Z^{2}}\overset{Z}{\underset{z=1}{\sum }}Var\left[\chi_{D_{f}}\left(X^{\left(z\right)}\right)\right]=\frac{1}{Z^{2}}ZP_{f}\left(1-P_{f}\right)=\frac{1}{Z}P_{f}\left(1-P_{f}\right)$$

Finally, we may write the dependability restriction, (β) [26,27] as:(17)$$\beta_{target}-\beta_{calc}\leq 0$$
where $\beta_{calc}$ is the calculated reliability index at the end of each iteration, and $\beta_{target}$ is the desired number over which the procedure must stop.

The following representations were employed in order to find out $\beta_{target}$ and $\beta_{calc}$:(18)$$\beta_{target}=-\Phi^{-1}\left(P_{f,target}\right)$$
(19)$$\beta_{calc}=-\Phi^{-1}\left(P_{f,calc}\right)$$

Thus, $\Phi^{-1}$ represents the inverse of the normal distribution function, which is the truncated normal distribution. Two of the concrete characteristics considered are compressive strength ${\overset{´}{f}}_{c\;}$ and modulus of elasticity $E_{c}$.

The code treats the reliability index as a limiting index that determined when the problem was solved by establishing a maximum value for the complementary strain energy, $W_{p}$; once this value was attained, the appropriate load, deflection, and complementary strain energy were computed.

### 2.3. Optimum Solution

The maximum plastic loading $F^{pl}$ that may be applied to the haunched beams without iterative updates to the constitutive components is determined via non-linear optimization. The deterministic solution takes into account the following equations, where $A_{i}$ and $l_{i}$ standing for the cross section and length of each element, respectively.
(20a)$$Max.\;\to \;F^{pl};$$
(20b)$$\;Subjected\;to:\;\;N^{el}=F^{-1}GK^{-1}P_{0};$$
(20c)$$-\bar{N^{pl}}\;\leq \;N^{Pl}\;\leq \;\bar{N^{pl}};$$
(20d)$$\frac{1}{2E}\;\overset{n}{\underset{i=1}{\sum }}\frac{l_{i}}{A_{i}}\;N_{i}^{R^{2}}\leq W_{p0}.$$
(20e)$$u-u_{o\;}<0.$$

The elastic fictitious internal normal forces are found in Equation (20b), and the lower and higher plastic limit conditions are presented in Equation (20c), where $N^{pl}$ is the ultimate plastic limit load. As an additional global metric of plastic behavior, boundary Equation (20d) displays the complementary strain energy of residual forces used to govern plastic deformations of steel bars.

In the meantime, the deflection condition is shown by Equation (20e), where u is the deflection value found by optimization, and $u_{o\;}$ is the maximum deflection value. For a deterministic solution, it is sufficient to assume that the plastic deformations are under control if the calculated value of the complementary strain energy of the residual forces is less than or equal to the bound for the magnitude of the allowable complementary strain energy of the residual forces, with the solution being terminated once the allowable complementary strain energy $\;W_{p0}$ is crossed.

However, in order to investigate the probabilistic solution, we substituted Equation (20d) with Equation (17) $\beta_{target}-\beta_{calc}\leq 0$, which states that the termination condition may vary from iteration to iteration based on the probabilistic complementary strain energy, with a mean value of ${\bar{W}}_{po\;}$and a standard deviation$\;\sigma_{w}$. If the calculated reliability index ($\beta_{calc}$) is higher than the allowed target value ($\beta_{target}$), the probabilistic solution is stopped. The study takes into account the optimization procedures depicted in Figure 1, although it is highlighted that the CDP parameters involved in the optimization problem do not change.

## 3. Model Validation 

In a recent study, Ibrahim and Rad [20] presented the models used in the current work. In that study, the dimensions of the strengthened concrete haunched beams shown in Figure 2 were as follows: dimensions of 2000 mm in length, 250 mm in depth at supports (hs), 150 mm in width, and 9 degrees in haunched angle (α). Steel-reinforcing bars of varying sizes have also been installed in the beams, as seen in Figure 3. 

Haunched beams’ non-linear behavior was modelled using finite element analysis, and the damage plasticity model (CDP) was used to represent concrete behavior in Abaqus. The concrete beam was modelled using the C3D8 element, a solid element with 8 nodes. The reinforcing bars were modelled as 2-node linear beam in space (B31) elements. The connection between longitudinal and transverse reinforcements in concrete was modelled using an embedded area. The carbon fiber-reinforced plastic (CFRP) was modelled as a 4-node (doubly curved shell) element (S4), and the bond between the CFRP and the concrete was modelled as a surface-to-surface contact, cohesive zone (glue), with a glue contact defined by a friction coefficient equal to 0.1 (which can be obtained from Abaqus manual [28]). In addition, two vertical concentrated loads were applied at each beam, with the load being dispersed via the coupling effect so that all experimental conditions were identical. As shown in Figure 4, after modelling the haunched beams, two strengthening instances were explored computationally and compared with experimental results. Case C2 indicates the occurrence of two U-wrap CFRP strips slanted by 45°, while case C3 represents the existence of three such strips. Figure 5 also depicted a comparison between the numerical and experimental results, revealing a good degree of consistency between the two sets of data.

To apply the reliability-based design while taking into account mean values of compressive strength ${\overset{´}{f}}_{c\;}$ = 35 MPa and a modulus of elasticity $E_{c}$ = 26,420 MPa, the probabilistic solution assigned a standard deviation of 5% to these concrete parameters, calculating the standard deviation of the complementary strain energy by taking its mean value across all deterministic cases (in line with strengthening scenarios) and using it as the probability estimation with a 10% standard deviation. In addition, the CFRP’s characteristics are subjected to probability analysis, with a 10% standard deviation taken into account, while considering the properties used inside Abaqus that are presented in Table 1 as mean values.

## 4. Results and Discussion

### 4.1. The Case of C2 Model with Two U-Wrap CFRP Strips

After passing through a numerical validation process, the results for the C2 model are presented in this section. The novel method was used to analyse strengthened haunched beams made of reinforced concrete, taking both deterministic and probabilistic approaches into account. Complementary strain energy $W_{po\;}$ was first used as a bound and took into account the deterministic solution (Equation (20d)). By establishing a probabilistic process and assuming the total sample point number (Z = 3 × 10^8^), the Monte Carlo approach was performed to compute the reliability indices, with the concrete properties, $W_{po\;}$, and CFRP properties modelled as random variables with mean and standard deviation. The provided reliability index can then be used as a bound $\beta_{target}$ (Equation (17)) in a probabilistic analysis. Both the deterministic and probabilistic methods relied on a check that the estimated deflection (u) was below the maximum allowed deflection value ($u_{o}$) before implementing the solution (20d).

Loads (*F*) and the corresponding deflection values (*u*) for each case are provided in Table 2, and the complementary strain energy ($W_{po}$) and reliability index ($\beta_{target}$) were chosen to regulate the deterministic and probabilistic analysis. The concrete characteristics are fluctuating at random within 5% mean value, and this would unquestionably alter the associated load values, demonstrating the uncertainty role. It can be seen that this table displays six possible outcomes. In cases (C2-0, C-2-0-1, and C2-0-2), a deterministic solution is displayed, where it can be seen that as complementary strain energy decreases in value, the produced load and deflection values also reduced. If we plot this in Figure 5, we can figure out that the produced deflection is similar to the deflection values in the figure which corresponds the same load values, and this proves the applicability of the used method and model. However, in cases (C2-1, C2-2, and C2-3), a variety of probabilistic solutions are displayed. Standard deviation $(\sigma_{w}$ = 10%) was applied to the mean value $({\bar{W}}_{po}$ = 120 N·mm), which was derived from the deterministic solution (C2-0), and a target reliability index ($\beta_{target}$) was used to determine three distinct probabilistic cases (C2-1 = 3.3, C2-2 = 4.9, C2-3 = 3.8). The properties of CFRP are also subjected to probabilistic analysis, with a 10% standard deviation taken into account. It can be seen from Table 2 that the randomness of the presented variables affects the load and deflection values clearly. Moreover, it can be seen that the reliability index value increases as the corresponding load value decreases, which reflects the role of increased probability of failure in the models subjected to higher loading conditions. 

Additionally, Table 3 displays other results showing the tension concrete damage percentages *d_t_* % initiated in each case, and these percentages are reflected in the damage pattern presented in the same table where high damage intensity is displayed by red areas while undamaged areas are displayed by blue colour. When the produced load is increased, the intensity of the damage also increases, declaring that the additional load applies additional initial stresses within the steel and concrete materials. This exemplifies the role of the complementary strain energy by reflecting the plastic damage, and generally, if this role is controlled, the plastic damage and failure are also controlled. 

### 4.2. The Case of C3 Model with Three U-Wrap CFRP Strips

This section held the same procedure explained in the previous section, but the model considered here has three CFRP strips instead of two. The method is used in the C3 model, with both deterministic and probabilistic outcomes taken into account after verifying that the computed deflection (*u*) was less than the maximum allowed deflection value ($u_{o}$), as included in Equation (20d).

Randomness in the properties of the variables, along with the complementary strain energy ($W_{po}$) and reliability index $(\beta_{target}$) used to regulate the deterministic and probabilistic analyses, respectively, affected the results shown in Table 4. It is evident that the concrete characteristics are fluctuating at random within a range of 5 percent of the mean values, and that this would unquestionably have an effect on the corresponding load values, demonstrating the uncertainty role. The table displays six possible outcomes. Different probabilistic solutions are shown for examples (C3-1, C3-2, and C3-3), with cases (C3-0, C3-0-1, and C3-0-2) being the deterministic answer where the complementary strain energy worked as bound, as it can be seen that the produced load and deflection values decrease as the inserted complementary strain energy decreases in value, and if we plot this in Figure 5, we can figure out that the produced deflection is similar to the deflection values in the figure which correspond to the same load values, proving the applicability of the used method and model. Every probabilistic case was assigned one of three values for the complementary strain energy $W_{po\;}$considering a standard deviation $(\sigma_{w}$ = 10%) on the mean value (${\bar{W}}_{po}$ = 90 N·mm) retrieved from the deterministic solution, and a target reliability index$\;(\beta_{target}$) in three different values (C3-1 = 4.7, C3-2 = 3.1, C3-3 = 4.3). CFRP features are also subjected to probabilistic analysis, with a 10% standard deviation taken into account. It can be seen from Table 4 that the randomness of the presented variables affecting the load and deflection values clearly and the reliability index increases as the corresponding load value is decreasing, declaring that smaller loads lead to more reliable solutions. That is how the reliability index works as a limitation.

Table 5 also shows the percentage of tension concrete damage, in *d_t_* %, that was initiated in each scenario; this percentage is represented in the damage pattern provided in the same table, with areas of high damage intensity shown in red and areas without damage shown in blue. By reflecting the plastic damage, the complementary strain energy role is highlighted, and if this is controlled, the plastic damage and failure are controlled as well. This is because an increase in the produced load causes an increase in the damage intensity, proving that the extra load applies additional initial stresses within the steel and concrete materials. The effectiveness of CFRP strips in absorbing the extra stresses caused by increasing applied loads can also be seen by comparing Table 3 and Table 5, where it is clear that the presence of the additional CFRP strips reduced the damaged red areas, even though the corresponding models’ strengths are increased.

## 5. Conclusions

To apply the deterministic optimization problem to the numerically calibrated haunched beams, this project compiled models of haunched beams made of CFRP-reinforced concrete, taking into account the complementary strain energy of the internal stresses initiated within the rebar and the reliability index as constraint factors. Similar haunched beams are also optimized using a probabilistic solution, which takes into consideration the concrete characteristics, complementary strain energy, and CFRP strips properties as probabilistic values with a mean value and standard deviation due to the fact that the reliability index is used as the basis for the proposed method’s consideration of probabilistic parameters for models with constrained plasticity. From these findings, we can draw the following inferences:It can be seen from the results that the randomness of the presented variables affected the load and deflection values clearly.The reliability index value increases as the corresponding load value decreases, and that reflects the role of increased probability of failure in the models subjected to higher loading conditions.Tension concrete damage percentages *d_t_* % are reflected in the damage pattern presented in the results, where it can be concluded that as the produced load increase the damage intensity also increases, declaring that extra load applies extra initial stresses inside steel and concrete materials. This is how the complementary strain energy role is highlighted by reflecting the plastic damage, and as it is controlled, the plastic damage and failure will also be controlled.The effectiveness of CFRP strips in absorbing the extra stresses caused by increasing applied loads can also be seen by comparing C2 and C3 results, where it is clear that the presence of the additional CFRP strips reduced the damaged areas even though the corresponding models’ strengths are increased.This research accounts for the concrete material’s uncertainties, which is its true state, since it is hard to determine its properties without probability. Since CFRP is one of the most effective materials for strengthening structural elements, the uncertainties of its characteristics were considered to explore how these probabilities affect the behaviour of reinforced concrete beams enhanced with varied numbers of CFRP strips with variable properties.It can be realised that in the deterministic cases, as the complementary strain, energy is spotted with different values, and the corresponding load and deflection values are changing. In addition, if it is taken into consideration, the complementary strain values increase significantly when approaching the ultimate load where plastic behaviour controls, and thus initiates, higher loading and deflection values.

## Figures and Tables

**Figure 1 polymers-15-00569-f001:**
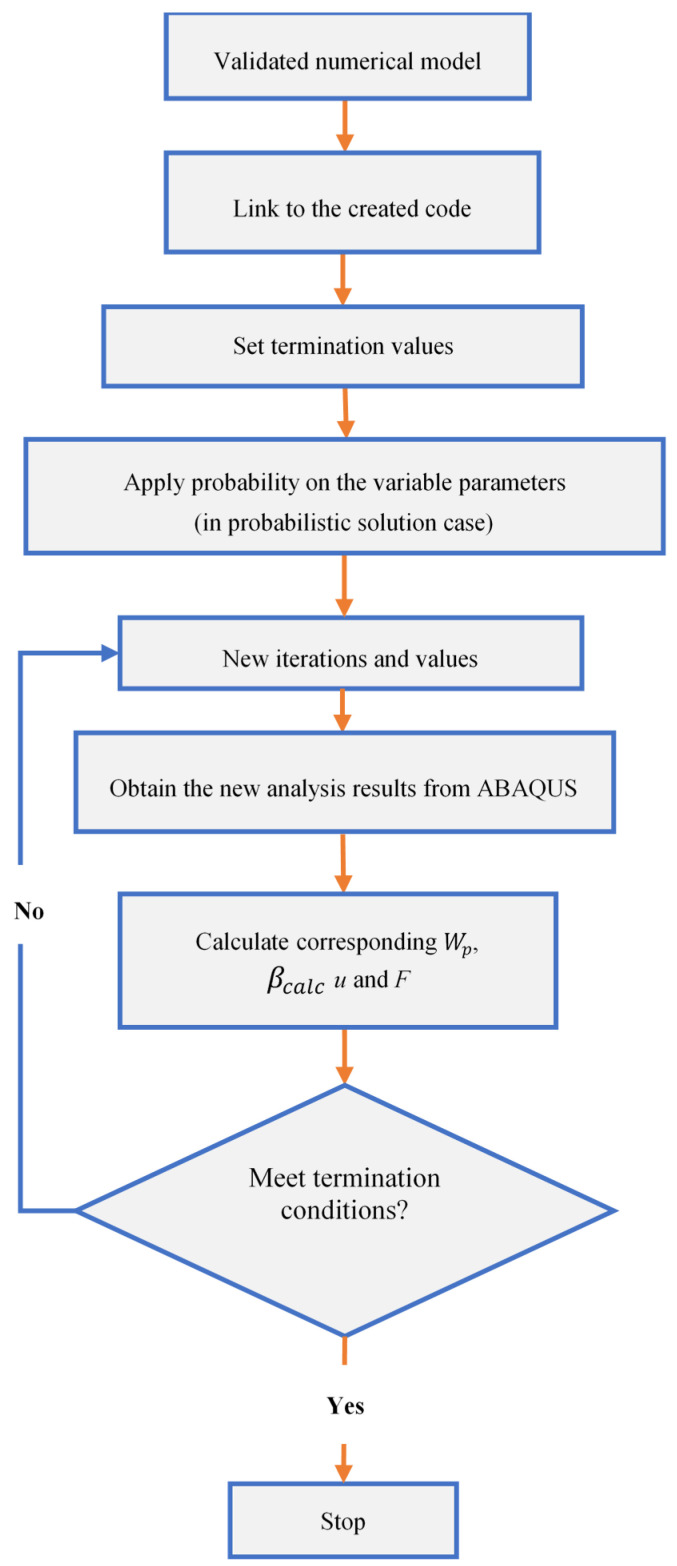
Optimization problem details.

**Figure 2 polymers-15-00569-f002:**
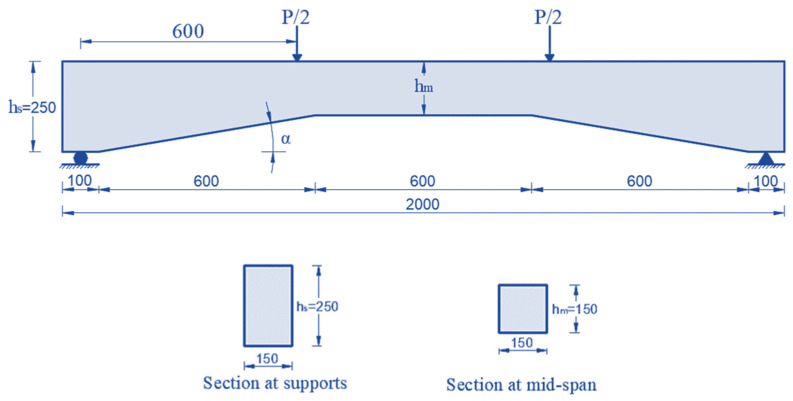
Haunched beams properties.

**Figure 3 polymers-15-00569-f003:**
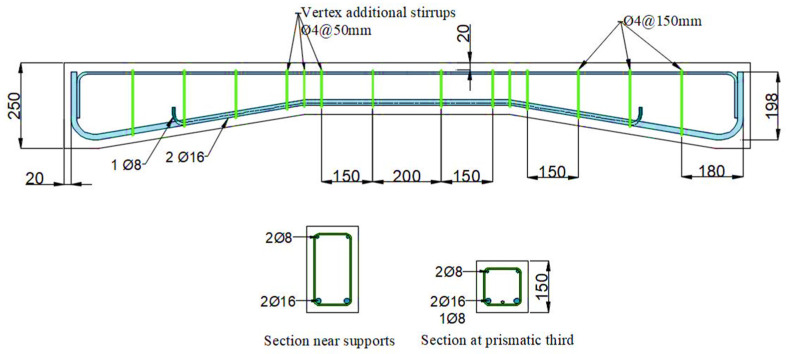
Rebar details for the used haunched beams.

**Figure 4 polymers-15-00569-f004:**
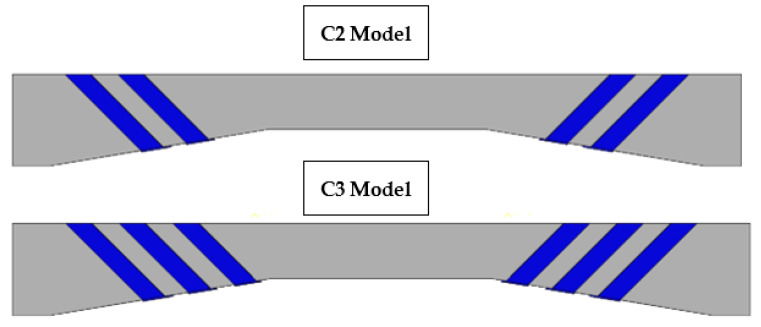
CFRP distribution for the haunched beams.

**Figure 5 polymers-15-00569-f005:**
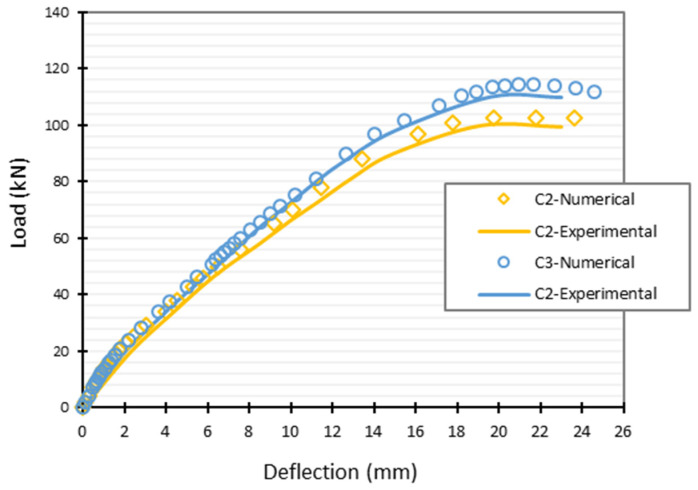
Numerical and experimental results comparison.

**Table 1 polymers-15-00569-t001:** CFRP properties.

E11 (MPa)	E22 (MPa)	Nu12	G12 (MPa)	G13 (MPa)	G23 (MPa)
140,000	10,000	0.26	5200	5200	3500

**Table 2 polymers-15-00569-t002:** Results obtained from C2 model.

Case		$W_{p0}$ (N·mm)	$\beta_{target}$	${\overset{´}{f}}_{c}$ (MPa)	$E_{c}$ (MPa)	*F* (kN)	*u* (mm)
Deterministic	C2-0	120	-	35	26,420	103	20
	C2-0-1	98				95	17.2
	C2-0-2	27				80	14.4
Probabilistic	C2-1	Randomly changed by 10%	3.1	Randomly changed by 5%		95	17
	C2-2		3.5			82	15
	C2-3		4.8			79	14

**Table 3 polymers-15-00569-t003:** Damage representation for C2 model.

Case	*d_t_* %	Tension Damage Pattern
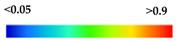
C2-0	31	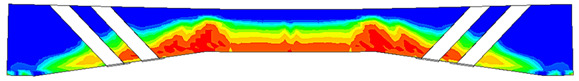
C2-1	29	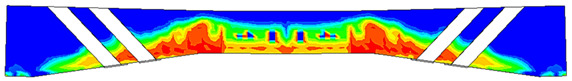
C2-2	28	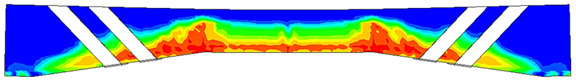
C2-3	26	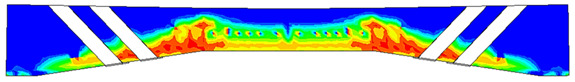

**Table 4 polymers-15-00569-t004:** Results obtained from C3 model.

**Case**		$W_{p0}$ (N·mm)	$\beta_{target}$	${\overset{´}{f}}_{c}$ (MPa)	$E_{c}$ (MPa)	*F* (kN)	*u* (mm)
Deterministic	C3-0	90	-	35	26,420	110	23
	C3-0-1	66				91	13
	C3-0-2	21				80	11.3
Probabilistic	C3-1	Randomly changed by 10%	3.1	Randomly changed by 5%		104	21
	C3-2		3.6			93	19
	C3-3		4.9			82	16

**Table 5 polymers-15-00569-t005:** Damage representation for C3 model.

Case	*d_t_* %	Tension Damage Pattern
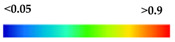
C3-0	27	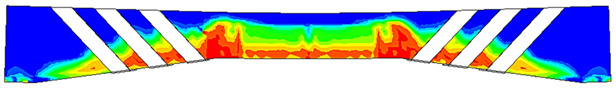
C3-1	25	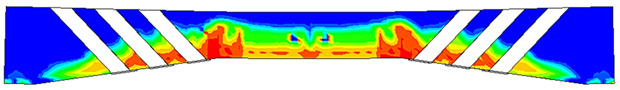
C3-2	23	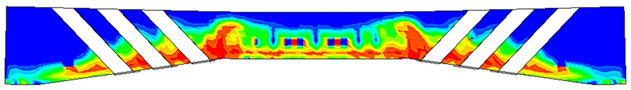
C3-3	21	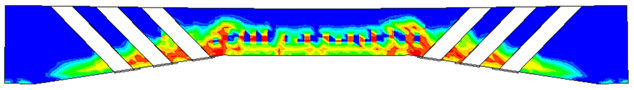

## Data Availability

The datasets which are generated during and analyzed the current study are available in the main manuscript, any additional details can be obtained from the Authors.

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
