# Peer review of "Limited Optimal Plastic Behavior of RC Beams Strengthened by Carbon Fiber Polymers Using Reliability-Based Design"

_polymers, 2023, doi:10.3390/polym15030569_

Round 1
Reviewer 1 Report
In this paper, the finite element simulation method is used to study the bearing capacity of concrete beams strengthened with CFRP, and the model is further verified by the relevant literature experimental results. It is suggested to consider the following comments to supplement some necessary information. In addition, the authors should enrich and improve the current paper by add the relevant analysis.
1. In the abstract, the authors should consider adding more qualitative and quantitative results and important findings. In addition, the innovation of the evaluation method proposed in this paper should be further highlighted.
2. Introduction, as an important reinforcement material for concrete beams in this paper, the information on CFRP is missing, such as performance, advantages and engineering applications compared to the metal materials and other FRPs. For example, the light weight, high strength, excellent corrosion resistance, fatigue resistance and creep resistance of CFRP are the key factors for strengthening concrete structures. The authors are suggested to make a necessary supplement through reviewing the following latest research work. Composite Structures 293, 115719. Composite Structures, 2019, 212: 243-258. Polymer Composites 41 (12), 5143-5155.
3. In the last paragraph of the introduction, it is suggested that the authors briefly highlight the main problems to be solved by the current research work and the relevant contributions related to CFRP strengthened RC beams. For example, how to use the research results of this paper to carry out the bearing capacity of concrete beams?
4. In the derivation of the model in Part 2, what are the limitations and applicability of the model?
5. What is the degree of agreement between the numerical simulation and the experimental results in Figure 5? In addition, the model proposed in this paper is only verified by the experimental results in Reference 13. Does this indicate that the current model has good applicability?
6. In part 4, the relevant simulation results (Table 2-5) are also recommended to be compared with the experimental results in the relevant literature to analyze the stability and accuracy of the current model.
7. After the accuracy of the current model is further verified, it is suggested that the authors further provide more simulation analysis results.
8. The conclusion part should be further improved, only including important qualitative and quantitative results and findings related to this paper.
Reviewer 2 Report
In this paper, the plastic behaviour of strengthened haunched beams using carbon fiber-reinforced polymers was investigated using a probabilistic design in order to use the reliability limitation index for simulated strengthened haunched beams by taking into account randomness in concrete and CFRP properties. The introduction of this paper may be improved by adding new bibliographic sources that present more research results within this field. The results indicate how randomness affects the behaviour of the presented models, which have different numbers of CFRP strips. In this regard, haunched beams were optimized using a probabilistic solution, which takes the concrete characteristics into consideration. Besides the FEA simulation, I think the samples should also be subjected to fire testing and mechanical tests.
After analysing the research objectives and the results, the reviewer considers that the paper should be improved by adding new bibliographic sources and performing experimental tests in order to be published in the journal Polymers.
Round 2
Reviewer 1 Report
The authors have well responded the comments.
Reviewer 2 Report
The introduction of this paper was improved by adding four new bibliographic sources, but I consider that there are still not enough.
Besides the FEA simulation, I consider that the samples should also be subjected to some experimental tests.
